# Association Between Muscle Quality Assessed by the 5-Repetition Sit-to-Stand Test and Falls in Community-Dwelling Older Adults in Japan: A Cross-Sectional Study

**DOI:** 10.3390/geriatrics10030078

**Published:** 2025-06-07

**Authors:** Koji Takimoto, Hideaki Takebayashi, Hiroshi Kondo, Koji Ikeda

**Affiliations:** 1Department of Rehabilitation, Faculty of Health Sciences, Naragakuen University, 3-15-1 Nakatomigaoka, Nara 631-8524, Japan; koji-ikeda@naragakuen-u.jp; 2Department of Rehabilitation, Faculty of Health Sciences, University of Kochi Health Sciences, 2500-2 Otsu, Kochi 781-5103, Japan; takebayashi@ko-ken-k3.ac.jp (H.T.); kondo@ko-ken-k3.ac.jp (H.K.)

**Keywords:** skeletal muscle mass (SMM), muscle quality index (MQI), 5-repetition sit-to-stand test (5R-STS), fall, community-dwelling older adults

## Abstract

**Background:** Falls in older adults are a major barrier to healthy longevity. Recent studies suggest that muscle quality is associated with fall risk. This study aimed to determine whether a functional muscle quality index (MQI) using the 5-repetition sit-to-stand test (5R-STS) reflects fall risk in community-dwelling older adults. **Methods:** This cross-sectional study included 137 community-dwelling older adults (≥65 years) in Japan. Lower limb skeletal muscle mass (SMM) was measured using the BIA method, and muscle function was assessed using the 5R-STS. The MQI was calculated as “(5R-STS (s)/SMM (kg)) × 10”. Fall history was collected using a self-administered questionnaire, and binary logistic regression analysis including gait speed and physical frailty was performed. **Results:** Participants were divided into fallers (n = 36; age = 78.2 ± 5.6) and non-fallers (n = 101; age = 76.9 ± 5.3). Significant differences were found between the groups in gait speed (*p* = 0.01), TUG (*p* < 0.01), 5R-STS (*p* < 0.01), and MQI (*p* < 0.01). Binary logistic regression identified MQI (OR = 1.28; *p* < 0.01) and gait speed (OR = 0.14; *p* = 0.02) as explanatory variables for fall history. The results of the evaluation using the receiver operating characteristic (ROC) curve showed that the cutoff value for MQI to distinguish fall history was 8.04 s/kg, and the cutoff value for gait speed was 1.21 s. **Conclusions:** The MQI using the 5R-STS shows promise as an indicator of fall risk in older adults. Further longitudinal studies are needed to clarify the causal relationship.

## 1. Introduction

Falls are a major threat to older adults worldwide, with 26.5% of people over 60 experiencing at least one fall [1]. In Japan, about 20% of community-dwelling older adults experience at least one fall per year [2,3], and recent reports indicate that 61.7% have experienced at least one fall in the past three years [4]. Falls can result in fractures such as hip fractures, leading to impaired functional abilities and restrictions on independent living. Therefore, preventing falls is an important challenge that must be addressed in an aging society to achieve healthy longevity.

On the other hand, muscle strength and muscle mass decrease with age and are known to be reflected in various health indicators in older adults [5,6,7]. The changes in skeletal muscle associated with aging are thought to be caused by various factors, including the infiltration of connective tissue and fat tissue into skeletal muscle, reduced protein synthesis, and a decrease in the number of motor units [8,9]. Since such changes in skeletal muscle significantly influence health indicators such as falls and fractures in older adults, they have recently garnered attention [10]. As demonstrated in previous studies, the decline in muscle strength in older adults is greater than that of muscle mass reduction [11,12]. This finding suggests the possibility of a decline in skeletal muscle quality.

It has been previously reported that muscle quality is associated with the risk of falls in older adults. For example, Yamada et al. assessed muscle mass (muscle thickness) and muscle quality (echo intensity) using ultrasound and reported that the incidence of falls increased with lower muscle quality and muscle mass [13]. In a recent study, Michel et al. used a torque machine to examine in detail the association with falls using the ratio of joint torque to skeletal muscle mass (SMM). The findings indicated that muscle quality is the most effective predictor of fall risk [14]. However, the utilization of ultrasound and torque machines has some problems, such as the need to use expensive specialized equipment and the requirement for a high level of expertise in its operation.

Skeletal muscle quality is generally described in two distinct domains: (1) the functional domain (e.g., muscle function per unit muscle mass) [15] and (2) the morphological domain (e.g., muscle architecture and composition) [16,17]. We believe that in the context of fall risk assessment in older adult individuals, the indicators of functional muscle quality, which reflect the strength and performance capacity of the lower limbs, are more sensitive to fall risk. Many studies have used the ratio of knee extensor strength to lean muscle mass as a measure of muscle quality assessment in older adult individuals [18]. On the other hand, a small number of reports have used sit-to-stand tests (e.g., 30 s chair stand test, 30-CS; 5-repetition sit-to-stand test, 5R-STS) as an alternative to knee extensor muscle strength [19,20,21]. The sit-to-stand test is known to be a simple indicator of lower limb muscle strength and to reflect the risk of falls [22,23,24], as it requires effort to quickly repeat standing and sitting. The 30-CS, which is one of the sit-to-stand tests, requires older adults to continue to stand and sit for 30 s with maximal effort, a challenge that is physically demanding and fatiguing. The 5R-STS, however, requires only five repetitions of standing and sitting, making it an assessment method with reduced burden and risk for older adults. Furthermore, the 5R-STS, together with the TUG and the Berg Balance Scale score, has been reported as the tool that most determines future fall risk in older adults [24]. Therefore, we believe that the utilization of the 5R-STS as a measure to assess fall risk in older adults is useful. The 5R-STS is also a performance test that requires movements of the lower limbs similar to standing and walking. Existing muscle quality assessments use the ratio of single joint torque to skeletal muscle mass as an indicator. We considered that using the ratio of performance ability to skeletal muscle mass as a muscle quality indicator would better reflect falls in older adults.

Indicators related to falls in older adults have been reported, including normal gait speed (m/s) [25], timed up and go test (TUG) (m/s) [26], handgrip strength (kg) [27], physical frailty [28], sarcopenia [10], and motoric cognitive risk syndrome (MCR) [29]. However, the extent to which the lower limb MQI (L/E MQI) using the 5R-STS reflects falls in older adults remains unknown. Consequently, it is also imperative to assess the extent to which the L/E MQI elucidates falls in older adults in comparison to established fall risk indicators.

The purpose of this study is to examine the relationship between the L/E MQI, which reflects the performance capacity per unit muscle mass of the lower limbs using the 5R-STS, and falls in older adults. By doing so, we will verify the extent to which the L/E MQI explains the history of falls in older adults, in conjunction with fall risk indicators such as gait speed and physical frailty, which are known to reflect fall risk.

## 2. Methods

### 2.1. Study Design

This study is a cross-sectional study conducted between September 2024 and March 2025 in Kōchi City, Kōchi Prefecture, and Nara City, Nara Prefecture, targeting community-dwelling older adults. This study has been approved by the Nara Gakuen University Ethics Committee (6-H012). Prior to the initiation of the study, informed consent was obtained from all participants.

### 2.2. Calculation of Sample Size

In this study, four explanatory variables were defined for binary logistic regression analysis. According to binary logistic regression principles, the minimum number of events (falls, in this case) required is determined by multiplying the number of explanatory variables by 10 [30]. Based on previous studies, the reported fall rate among community-dwelling older adults is approximately 30%. Assuming a sample size of 150 participants, 30% of this sample (n = 45) meets the requirement for the number of explanatory variables, thereby justifying the chosen sample size.

### 2.3. Participants

The participants in this study were community-dwelling older adults aged 65 years or older who participated in community-based exercise classes. These classes are known to be effective in preventing disability and are widely implemented throughout Japan [31]. The classes were held once a week, with each session lasting 60–90 min and consisting of group exercises (including stretching and strength training). The inclusion criteria for participants were age 65 or older and the ability to live independently in daily life. Exclusion criteria included those who had received Long-Term Care Insurance System certification with a care level of 1 or higher, those with a history of central nervous system or mental disorders, and those currently undergoing treatment and visiting a medical facility. It is important to note that this study did not conduct detailed cognitive function tests. However, individuals with cognitive impairment requiring assistance in daily living are classified as requiring long-term care under the Long-Term Care Insurance System. Consequently, the absence of long-term care certification was utilized as evidence of no cognitive impairment. A total of 3 of the 150 participants met the exclusion criteria. In addition, 10 participants had missing values on physical function tests or questionnaire responses. Finally, data from 137 participants were included in the statistical analysis (Figure 1).

### 2.4. Outcome Measure

The outcome measure of fall history was assessed by means of a self-administered questionnaire, in which participants were asked to report whether they had fallen in the past year. In this study, a fall was defined as an unintentional loss of balance resulting in the participant falling to a lower level than their original position.

Falls due to special environmental factors, such as traffic accidents or falls while riding a bicycle, were not included [32].

### 2.5. Calculation of MQI (Muscle Quality Index)

The muscle quality used in this study was evaluated using a macro-level indicator based on muscle function per unit of muscle mass. The measurement of muscle mass was conducted utilizing the InBody470 body composition analyzer (InBody Japan Inc., Tokyo, Japan), which employs bioelectrical impedance analysis (BIA). The InBody470 is capable of measuring muscle mass by body segment; therefore, the sum of the skeletal muscle mass of the lower limbs was extracted from the data.

The 5R-STS was adopted as an indicator of lower limb muscle function. The starting position of the participants was seated on a chair with a seat height of 40 cm. Subsequent to the signal from the examiner to commence, the participants were instructed to repeat the standing and sitting movements in sequence five times. The time required to complete the fifth standing movement was measured using a digital stopwatch. The 5R-STS was performed twice, and the fastest time (s) was adopted.

Using the SMM (kg) and 5R-STS (s) values obtained from the above measurements, the lower extremity (L/E) MQI was calculated using the following formula: L/E MQI (muscle quality index) = (5R-STS (s)/SMM (kg)) × 10. It is important to note that the values were multiplied by 10 in the calculation formula to adjust the scale. This is because the values of 5R-STS (s)/SMM (kg) tend to be small, resulting in a large impact of changes per unit and potentially leading to an inflated odds ratio.

### 2.6. Explanatory Variables

Various indicators reflecting fall risk in older adults have been identified to date. The L/E MQI, which is the focus of this study, needs to be evaluated to confirm whether it appropriately reflects fall risk in older adults compared to conventional fall risk indicators. Therefore, in addition to L/E MQI and 5R-STS, the following known fall risk indicators were evaluated as candidate explanatory variables: usual gait speed (m/s) [25], TUG (m/s) [26], grip strength (kg) [27], physical frailty [28], sarcopenia [10], and MCR [29].

Gait speed was calculated using on the time taken to walk a flat 5 m path at the usual pace. TUG was measured by timing the time required to stand up from a seated position, walk to a pole 3 m ahead, and return to the chair and sit down. Grip strength was measured using a grip strength dynamometer (GRIP-D, TAKEI Scientific Instruments Co., Ltd., Niigata, Japan), with two measurements taken on each hand, and the maximum value was adopted.

Physical frailty was assessed using the Japanese version of the CHS criteria (J-CHS criteria) to assess physical frailty, and classified participants into three categories: robust, pre-frail, and frail [33].

Sarcopenia is defined by the AWGS 2019 criteria as a condition in which a decrease in skeletal muscle mass is an absolute requirement [34]. However, individuals with reduced muscle strength or physical function but no reduction in SMM are excluded from the definition of sarcopenia. Therefore, the criteria proposed by Yamada et al., which classify individuals with reduced muscle strength or physical function but no reduction in SMM as dynapenia, were adopted to categorize participants into four groups: robust, dynapenia, pre-sarcopenia, and sarcopenia [35].

MCR was defined and determined based on the method of Verghese et al., as meeting two criteria: a decrease in walking speed (below one standard deviation of the mean walking speed of the target population) and a decline in subjective cognitive function (answering “yes” to the question “Do you feel you have more problems with memory than most?”) [36].

### 2.7. Covariate Variables

We also collected age and gender via the questionnaire to control for confounding factors affecting fall history.

### 2.8. Statistical Analysis

To examine differences in the MQI and other assessment indicators between the two groups based on fall history, normality was first assessed using the Shapiro–Wilk test. An independent t-test, Welch’s *t*-test, or the Mann–Whitney U test was then performed. For categorical indicators, the following numerical values were used: physical frailty (0: robust; 1: pre-frailty; 2: frailty), sarcopenia (0: robust; 1: pre-sarcopenia; 2: dynapenia; 3: sarcopenia), MCR (0: robust; 1: MCR). For physical frailty, sarcopenia, and MCR, the association with fall history was assessed using either the chi-squared test for independence or Fisher’s exact test.

We then performed binary logistic regression analysis with the indicators showing significant differences or associations between the presence or absence of a history of falls as explanatory variables and the presence or absence of a history of falls as the objective variable. Age and sex were treated as confounding factors. Variable selection was performed using the stepwise method. Subsequently, the discriminatory performance of variables reflecting fall history, as identified through binary logistic regression analysis, was evaluated through the construction of a receiver operating characteristic (ROC) curve. The ROC curve was utilized to determine the cutoff value for ascertaining the presence or absence of fall history, and the sensitivity and specificity were subsequently calculated. Furthermore, the discriminatory performance was evaluated using the area under the curve (AUC).

All statistical analyses were conducted at a significant level of 5%. Statistical analyses were performed using Bellcurve for Excel Ver 4.07 and R Ver 4.0.2 for Windows.

## 3. Results

### 3.1. Characteristics of Participants and Comparison of Fallers and Non-Fallers

Of the 137 subjects analyzed, 36 (26.3%) had a history of falls and 101 (73.3%) did not. Statistical analysis for differences in each outcome measure revealed significant differences in gait speed (*p* = 0.01), TUG (*p* < 0.01), 5R-STS (*p* < 0.01), and MQI (*p* < 0.01). In addition, physical frailty was found to be significantly associated with a history of falls (*p* = 0.01) (Table 1).

### 3.2. Results of Binary Logistic Regression Analysis

We performed a binary logistic regression analysis with indicators showing differences based on fall history (gait speed, TUG, 5R-STS, and MQI) and physical frailty associated with fall history as explanatory variables and fall history as the dependent variable. Sex and age were included in the analysis as confounding variables. The results showed that the coefficient of determination (R^2^) was 0.11 and the regression equation was significant (*p* < 0.01). The variables that explained fall history were MQI (OR = 1.28; *p* < 0.01) and gait speed (OR = 0.14; *p* = 0.02), which were selected as variables to be included in the regression equation (Table 2).

### 3.3. Discrimination Performance and Cutoff Value of Fall History Using the ROC Curve

The ROC curve was used to evaluate the discriminatory performance of MQI and gait speed for fall history. The discriminatory performance of MQI for fall history was evaluated, and the AUC was 0.67, the cutoff value was 8.04 (s/kg), the sensitivity was 0.61, and the specificity was 0.70. On the other hand, the discriminatory performance of gait speed for fall history was evaluated, and the AUC was 0.65, the cutoff value was 1.21 (s), the sensitivity was 0.64, and the specificity was 0.65 (Figure 2).

## 4. Discussion

This study investigated whether the MQI, calculated using the ratio of 5R-STS (s) to lower limb SMM (kg), reflects falls in community-dwelling older adults. The results showed that the MQI and gait speed are indicators that reflect fall history in older adults.

The fall rate of the participants in this study was 26.3%, which was generally consistent with the fall incidence rates reported in previous studies [2,3]. On the other hand, the participants in this study had shorter gait speeds and TUG times than those reported in previous studies [24,37]. In addition, the prevalence of frailty and sarcopenia in Japanese older adults [38,39] was lower among the participants in this study. Based on the above, it can be assumed that the participants in this study had a relatively high level of physical fitness levels even among community-dwelling older people in their 70s. Gait speed, TUG, 5R-STS, and physical frailty, which are known to be simple indicators of fall risk, showed significant differences between fallers and non-fallers. However, an interesting finding of this study was that the multivariate analysis revealed that the MQI using the 5R-STS, which was the focus of this study, was extracted as a variable explaining fall history along with gait speed.

Previous studies have examined the MQI using the ratio of SMM (kg) to muscle function [18]. Methods to assess the SMM include computed tomography (CT) and magnetic resonance imaging (MRI) [40], and dual-energy X-ray absorptiometry (DXA) [41]. However, these methods have limitations, such as exposure to X-rays, high equipment costs, and limited measurement opportunities. Ultrasound can measure skeletal muscle thickness and cross-sectional area with high reproducibility and reliability [42], but it requires advanced technical skills for measurement. On the other hand, the body composition analyzer device using BIA used in this study enables the rapid measurement of the SMM. Therefore, considering safety, burden, and measurement efficiency, we believe that the BIA method is the most appropriate method for measuring the SMM in older adults.

On the other hand, muscle strength has traditionally been used to assess lower limb muscle function. However, it has been reported that the muscle strength of a single muscle group alone is insufficient to predict future falls in older adults [43]. In contrast, the 5R-STS used in this study reflects not only simple lower limb muscle strength but also muscle power [44]. During standing or walking, postural and motor control under weight-bearing conditions require closed kinetic chain (CKC) movements of the lower limbs. Therefore, the 5R-STS is expected to better reflect actual standing balance and walking ability than single-joint torque measured in an open kinetic chain (OKC) condition. This is supported by numerous studies reporting that the 5R-STS reflects fall risk [44,45,46]. In this study, while the discrimination performance (AUC: 0.67) for determining a history of falls was not sufficiently high, a cutoff value of 8.04 (s/kg) for the L/E MQI was demonstrated, which may be useful as an indicator of fall risk. However, the implementation of prospective cohort studies is imperative to utilize this as a reliable indicator of fall risk. Therefore, we believe that the SMM measured by BIA, which can be measured safely and rapidly in community-dwelling older adults, and the MQI calculated from the 5R-STS, which sensitively reflects fall risk, are useful assessment tools that can be easily evaluated and are highly practical for reflecting fall risk.

One of the challenges of the MQI used in this study is that individuals with painful musculoskeletal conditions, such as low back pain or osteoarthritis of the lower limbs, may not be able to fully perform the 5R-STS. Of course, the presence of pain itself may be reflected in fall risk [47], but in such cases, it is necessary to supplement the results with other fall risk indicators obtained in this study, such as gait speed, the TUG, and the Berg Balance Scale, which are effective assessments of fall risk [24]. Furthermore, gender differences in muscle quality are known to exist, and studies using the ratio of knee flexion and extension torque to muscle cross-sectional area have reported that muscle quality is lower in women than in men [48]. In this study, about 80% of the participants were women, so it should be noted that the cutoff values for muscle quality indicators used to determine fall history may have been low.

### 4.1. Clinical Implications

The clinical implication of this study is that the L/E MQI using the 5R-STS was found to be useful in distinguishing between elderly individuals with and without a history of falls. Given the established link between age-related muscle function decline and falls, a novel approach was devised to assess muscle function in older adults. This method was designed to be both safe and expeditious, thereby minimizing the burden on these individuals. The finding that the L/E MQI is an effective indicator for determining fall history in combination with gait speed has practical significance.

### 4.2. Limitations

The limitations of this study include other factors related to falls (detailed cognitive function assessment, physical activity level, lower limb muscle strength, etc.) were not sufficiently investigated. In addition, because this is a cross-sectional study, the causal relationship between the MQI based on the 5R-STS and falls cannot be explained. Furthermore, selection bias may have been a factor because the participants in this study were community-based exercise class participants. Moreover, the proportion of female participants constituted about 80%. This demographic characteristic may have influenced the outcomes, potentially leading to lower muscle quality indices. The results of this study should be interpreted considering the aforementioned factors.

### 4.3. Future Studies

This study is only a pilot study investigating the association between the MQI by the 5R-STS and falls. Further prospective cohort studies are needed to establish causal relationships. Consequently, prospective cohort studies must be conducted to further verify the causal relationship between the L/E MQI and falls in order to establish practical indicators for fall prevention in older adults.

## 5. Conclusions

As people age, skeletal muscle quality declines, and it is known that muscle quality decline is associated with health indicators in the elderly, such as falls. In this study, we calculated the MQI from the ratio of 5R-STS (s) to lower limb SMM and examined whether it reflected falls in older adults. Consequently, gait speed (OR: 0.14; cutoff value: 1.21 s) and the L/E MQI using the 5R-STS (OR: 1.28; cutoff value: 8.04 s/kg) were selected as indicators for determining fall history in community-dwelling older adults. In order to utilize the L/E MQI as an indicator of fall risk that focuses on changes in muscle quality associated with aging, it is necessary to conduct a prospective cohort study to verify the causal relationship between the L/E MQI and falls.

## Figures and Tables

**Figure 1 geriatrics-10-00078-f001:**
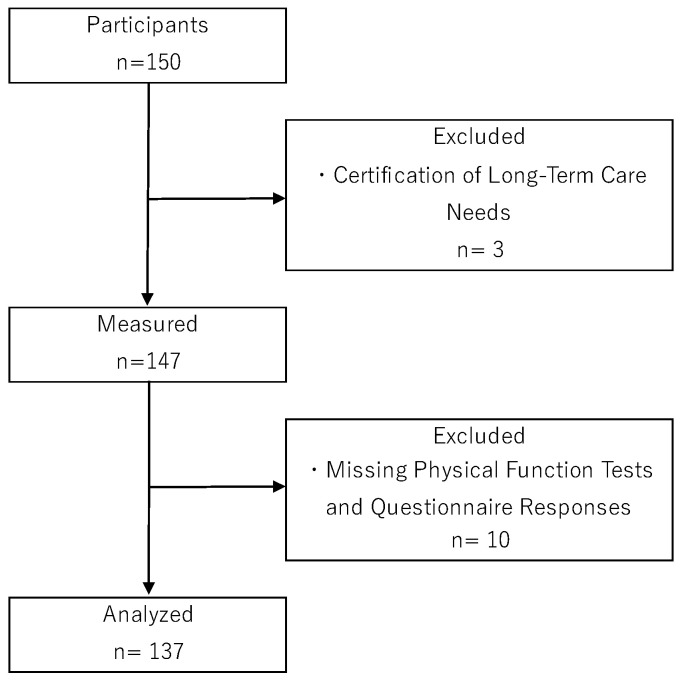
Flow chart of participant selection.

**Figure 2 geriatrics-10-00078-f002:**
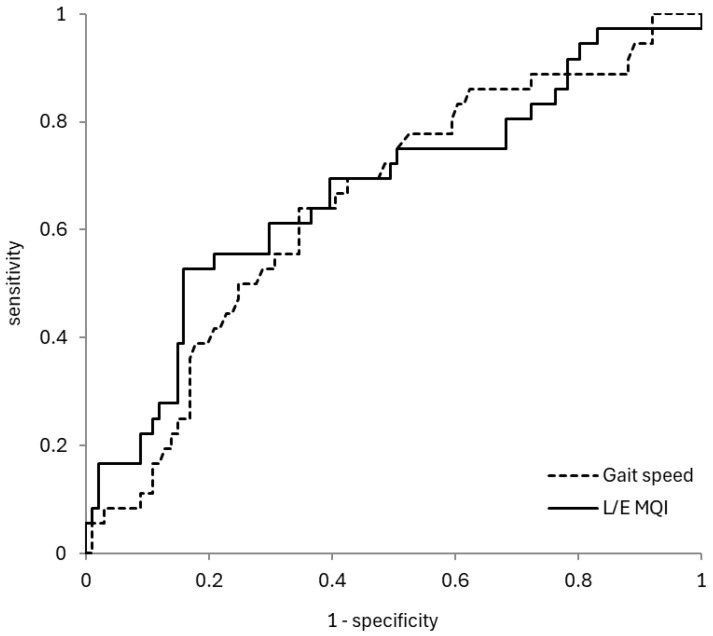
ROC curves of L/E MQI and gait speed for determining fall history. LE/MQI: AUC (0.67), cutoff value (8.04 s/kg), sensitivity (0.61), specificity (0.70). Gait speed: AUC (0.65), cutoff value (1.21 s), sensitivity (0.64), specificity (0.65). ROC: receiver operating characteristic; AUC: area under the curve; L/E MQI: lower extremity muscle quality index; TUG: timed up and go test.

**Table 1 geriatrics-10-00078-t001:** Characteristics of the participants and comparison of fallers and non-fallers.

Variable	Faller (n = 36, 26.3%)	Non-Faller (n = 101, 73.3%)	*p*-Value		Effect Size	
Age (years)	78.2 ± 5.6	76.9 ± 5.3	0.22	#	0.24	
Sex; women (n, %)	29 (80.6)	82 (81.2)	0.93	*	<0.01	^¶^
BMI (kg/m^2^)	22.7 ± 3.4	22.1 ± 3.0	0.33	#	0.19	
Gait Speed (m/s)	1.19 ± 0.24	1.32 ± 0.26	0.01	#	0.50	
TUG (s)	8.74 ± 2.12	7.76 ± 1.31	<0.01	###	0.63	
Grip Strength (kg)	22.8 ± 5.6	25.0 ± 5.9	0.07	###	0.37	
5R-STS (s)	9.7 ± 3.0	8.0 ± 2.3	<0.01	##	0.66	
SMI (kg/m^2^)	6.09 ± 0.64	6.23 ± 0.86	0.29	##	0.18	
Lower leg SMM (kg)	11.08 ± 1.90	11.64 ± 2.57	0.37	###	0.23	
Lower leg MQI (s/kg)	8.92 ± 3.18	7.13 ± 2.30	<0.01	###	0.70	
Physical Frailty (n, %) Robust Pre-Frail Frail	17 (12.4) 14 (10.2) 5 (3.6)	55 (40.1) 45 (32.8) 1 (0.7)	0.01	**	0.28	^¶¶^
Sarcopenia (n, %) Robust Pre-Sarcopenia Dynapenia Sarcopenia	18 (13.1) 6 (4.4) 8 (5.8) 4 (2.9)	62 (45.3) 23 (16.8) 10 (7.3) 6 (4.4)	0.17	**	0.19	^¶¶^
MCR (n, %) Non-MCR MCR	32 (23.4) 4 (2.9)	94 (68.6) 7 (5.1)	0.48	**	0.07	^¶^

Values are mean ± SD or n and proportion. #: unpaired *t*-test; ##: Welch’s *t*-test; ###: Mann–Whitney U test; *: chi-squared test; **: Fisher’s exact test. Effect size: ^¶^: phi coefficient; ^¶¶^: Cramer’s V; others: Cohen’s d. BMI: Body Mass Index; TUG: timed up and go test; 5R-STS: 5-repetition sit-to-stand test; SMI: skeletal muscle mass index; SMM: skeletal muscle mass; MQI: muscle quality index; MCR: motoric cognitive risk syndrome.

**Table 2 geriatrics-10-00078-t002:** Association of falls with MQI and other variables.

	Odds Ratio	95%CI [Min–Max]	Wald	*p*-Value
Gait Speed	0.14	[0.03–0.76]	5.17	0.02
L/E MQI	1.28	[1.09–1.51]	9.46	<0.01

Adjusted R^2^: 0.11; objective variable: faller/non-faller; explanatory variables: gait speed, timed up and go test, 5-repetition sit-to-stand test, L/E MQI, physical frailty, age, and sex. CI: confidence interval; L/E MQI: lower extremity muscle quality index.

## Data Availability

The data supporting the findings of this study are available upon request from the corresponding author. The data are not publicly available because of privacy and ethical restrictions.

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
