# Peer review of "Association Between Muscle Quality Assessed by the 5-Repetition Sit-to-Stand Test and Falls in Community-Dwelling Older Adults in Japan: A Cross-Sectional Study"

_geriatrics, 2025, doi:10.3390/geriatrics10030078_

Round 1
Reviewer 1 Report
Comments and Suggestions for Authors
Dear Authors,
I consider that your research is updated and important, as one of the most significant social shifts we have been witnessing during the last decades is the aging of the population. Scientific data reported that in 2020 the number of adults 65 and over were estimated at 723.484 million and is expected to reach 1.5 billion by 2050. This phenomenon shows that there is an increased need for more research for improving the medical care programs and quality of life for elderly people.
In the Abstract please reformulate the results, explaining with words what the statistical analysis revealed, as it would be more easily to understand by clinicians who are not very familiar with statistical tests. Based on the conclusion here you showed that MQI using 5R-STS could be an indicator for fall risk, but you give no cut-off values, which affects the clinical relevance of the study.
The Introduction is very long, with some information that would better be placed in the Discussion (for example, lines 47-49). However, I consider that you provided enough background for the research, in a logical and synthetic manner, easy to follow and to understand. In the end you formulate a hypothesis, but throughout the manuscript you did not follow-up and explain whether it was confirmed or not.
In Methods there is an accurate description of the study methodology and variables used, but there are many explanations and references that do not belong here (ex. Lines 159-160, 181-187). Please explain why you decided to measure other six variables (section 2.6).
The Results reflect all the investigations performed and are correctly evaluated based on the used statistical tests.
In Discussion please explain whether the fact that the study group includes 81% women could affect the results, as you make many comparisons with other studies related to „older adults”. Even though it is only a pilot study, I expected a clear statement regarding the possibility to use the 5R-STS as an indicator for the risk of falls in elderly and a range of values that might be suggestive for such an event.
In my opinion the Conclusions must be reformulated, they should reflect the authors' opinion about the importance of their research and how it could be continued and implemented, in order to gain clinical relevance.
Author Response
We sincerely thank you for your thought-provoking advice on our submitted paper. We have responded to your comments as follows, which you can review along with the revised manuscript.
Comment 1: In the Abstract please reformulate the results, explaining with words what the statistical analysis revealed, as it would be more easily to understand by clinicians who are not very familiar with statistical tests. Based on the conclusion here you showed that MQI using 5R-STS could be an indicator for fall risk, but you give no cut-off values, which affects the clinical relevance of the study.
Response 1: Thank you for pointing out the abstract. We added the MQI cutoff value based on the ROC curve to the results, which conveys the clinical implications of this study. This addition will make the statistical analysis results and clinical significance easier for readers to understand. Please refer to P1, L25–27.
Comment 2: The Introduction is very long, with some information that would better be placed in the Discussion (for example, lines 47-49). However, I consider that you provided enough background for the research, in a logical and synthetic manner, easy to follow and to understand. In the end you formulate a hypothesis, but throughout the manuscript you did not follow-up and explain whether it was confirmed or not.
Response 2: Thank you for your comments about the Introduction. In this study, we thought it was important to show that muscle quality is related to the risk of falling, and we mentioned this in the introduction. However, to keep the introduction from being too long, we deleted the sentence you mentioned and shortened the paragraph (P2, L50-58). Please check out the updated version. However, please note that the Introduction is not significantly shorter than before. This is because other reviewers gave instructions about the hypothesis presented in the Introduction.
Regarding the response to the hypothesis, the additional information has been incorporated into the newly added sections 4.1. Clinical implications (P9, L331-338) or 5. Conclusions (P9, L358-367). Please review these sections. (However, the explanation of the hypothesis has been revised based on comments from other reviewers.)
Comment 3: In Methods there is an accurate description of the study methodology and variables used, but there are many explanations and references that do not belong here (ex. Lines 159-160, 181-187). Please explain why you decided to measure other six variables (section 2.6).
Response 3: Thank you for your comments regarding the Methods.
First, in 2.5 Calculation of MQI (P4, L157-), we cited two pieces of evidence reporting the reliability and validity of the 5R-STS. We included these because the 5R-STS is important for this study, but we have deleted them in response to your suggestion that they are unnecessary.
Second, we will explain why we measured the other six variables. There are different ways to measure the risk of falls in older adults. This study talks about six of these ways. So, we checked how well the MQI, the main focus of this study, could spot fall risk in older adults compared to these indicators. This point has been added to Section 2.6 (P5, L179-185) along with the Introduction (P2, L82-88). Please review these sections.
Comment 4: The Results reflect all the investigations performed and are correctly evaluated based on the used statistical tests.
Response 4: Thank you very much for carefully checking the results.
Comment 5: In Discussion please explain whether the fact that the study group includes 81% women could affect the results, as you make many comparisons with other studies related to „older adults”. Even though it is only a pilot study, I expected a clear statement regarding the possibility to use the 5R-STS as an indicator for the risk of falls in elderly and a range of values that might be suggestive for such an event.
Response 5: Thank you for your important comment. Since most of the participants in this study were women, it is highly possible that gender affected the results. We appreciate your comment, as gender differences in muscle quality are an important point to mention. Please refer to the additions made to the Discussion (P9, L324-329) and Limitations of the study (P9, L346-349).
We also added an analysis using a ROC curve in the revised manuscript. This allowed us to clearly show how applicable the results are by indicating cutoff values. Regarding the ROC curve, we described the method (P5, L225–231), results (P7, L263–274, and Figure 2), and discussion (P8, L311–314) in the revised manuscript. 4.1. Clinical implications (P9, L331-338) include a description of usability, so please review these sections.
Comment 6: In my opinion the Conclusions must be reformulated, they should reflect the authors' opinion about the importance of their research and how it could be continued and implemented, in order to gain clinical relevance.
Response 6: Thank you for your comments on the structure of the Conclusion. As you pointed out, we have restructured the conclusion to emphasize the importance and clinical implications of this study and the need for further research. Please refer to P9, L359-367.
This is our response to the comments. We appreciate the helpful comments on our paper.
Reviewer 2 Report
Comments and Suggestions for Authors
Dear all, here are some suggestions to improve your study:
Abstract
It is well written and meets the requirements of a summary.
Introduction
1. throughout the text, avoid using the word "elderly" (ageism). I suggest replacing it with "older adult";
2. I have strong doubts about the expression, "In this study, it was hypothesized that a functional muscle quality index (IMQ) using the 5R-STS, a dynamic performance measure, would more accurately reflect the risk of falls in the elderly" is not complete! In particular, the excerpt "would more accurately reflect the risk of falls". Question: Would reflect more accurately than what? What is the comparison? There is no other test/assessment to be compared. Therefore, the theoretical framework of the hypothesis is weak/flawed! I suggest improving it...
3. The objective of the study was well formulated. However, see my comments in the "methodology" section: the methodological basis is confusing because the authors associated other indicators and covariates in section 2.6. This makes the study confusing. Covariates are variables that are estimated to affect the results, acting on the dependent variable (falls). This is not clear in this section. Therefore, I suggest separating and presenting "covariates" in a single section;
4. The formulation of the previous hypothesis, in addition to being confusing, is still precarious in its present form because the authors have a series of other indicators of falls that were not mentioned in the objective. Therefore, they should be presented as hypotheses. Therefore, I suggest the formulation of hypotheses for "other indicators related to falls, including usual walking speed, (m/s) [31], Timed Up and Go test; TUG (m/s) [32], grip strength (kg) [33], physical frailty, sarcopenia, and motoric cognitive risk syndrome (MCR). If this does not occur, the intentions of the study and presentation of the results will remain unharmonious [the methodological basis will be weak/confusing].
Methods
1. The section has problems, some of which have already been enumerated;
2. This sentence "We also collected age and gender via questionnaire to control for confounding factors affecting fall history" is missing at the end of the text;
3. In section "2.7. Statistical Analysis" there is a lack of description of the treatment of categorical data;
4. The presentation of the analysis procedures contains "logistic regression", however, in the summary, there is an indication of "multivariate analysis". Therefore, confusing: all of this should be rigorously reviewed.
Results
1. Again, something confusing: "3.2. Results of binary logistic regression analysis". Where are the results of the multivariate analysis?
Discussion
1. At the end of this section, there should be a subsection titled "Clinical Implications." In this subsection, the authors should have presented extensive recommendations for practitioners on how to use their findings in fall prevention;
2. Therefore, I suggest creating another subsection titled "Strengths, Limitations, and Future Studies"
Author Response
We sincerely thank you for your thought-provoking advice on our submitted paper. We have responded to your comments as follows, which you can review along with the revised manuscript.
Introduction
Comment 1: throughout the text, avoid using the word "elderly" (ageism). I suggest replacing it with "older adult";
Response 1: We apologize for using inappropriate terminology. We have replaced all instances of the term “elderly” in this paper with “older adults.”
Comment 2: I have strong doubts about the expression, "In this study, it was hypothesized that a functional muscle quality index (IMQ) using the 5R-STS, a dynamic performance measure, would more accurately reflect the risk of falls in the elderly" is not complete! In particular, the excerpt "would more accurately reflect the risk of falls". Question: Would reflect more accurately than what? What is the comparison? There is no other test/assessment to be compared. Therefore, the theoretical framework of the hypothesis is weak/flawed! I suggest improving it...
Response 2: Thank you for pointing out our hypothesis. Our explanation of the hypothesis wasn't good enough. In this study, we attempted to explain the history of falls using the conventional indicators of fall risk (gait speed, TUG, grip strength, physical frailty, sarcopenia, and MCR) and the newly proposed indicator, L/E QMI, as explanatory variables. At that time, we hypothesized that L/E MQI might have greater explanatory power (higher discriminative performance) for fall history compared to the other indicators adopted in this study. We acknowledge that the rationale for this hypothesis was insufficiently explained. Therefore, we have revised the section outlining the hypothesis in the Introduction. Please refer to P2, L77–88 for the revised text.
Comment 3: The objective of the study was well formulated. However, see my comments in the "methodology" section: the methodological basis is confusing because the authors associated other indicators and covariates in section 2.6. This makes the study confusing. Covariates are variables that are estimated to affect the results, acting on the dependent variable (falls). This is not clear in this section. Therefore, I suggest separating and presenting "covariates" in a single section;
Response 3: Thank you for your feedback. We have indicated the corrections and the locations where they have been made in the "Comment 5" section of the "Methods" below. Please refer to that section for details.
Comment 4: The formulation of the previous hypothesis, in addition to being confusing, is still precarious in its present form because the authors have a series of other indicators of falls that were not mentioned in the objective. Therefore, they should be presented as hypotheses. Therefore, I suggest the formulation of hypotheses for "other indicators related to falls, including usual walking speed, (m/s) [31], Timed Up and Go test; TUG (m/s) [32], grip strength (kg) [33], physical frailty, sarcopenia, and motoric cognitive risk syndrome (MCR). If this does not occur, the intentions of the study and presentation of the results will remain unharmonious [the methodological basis will be weak/confusing].
Response 4: Thank you for your important comment. We have added the reasons for using other indicators related to falls, as you specifically pointed out, to the Introduction, including the variable names. As mentioned in Comment 1 above, please refer to P2, L82-88.
Methods
Comment 5: The section has problems, some of which have already been enumerated;
Response 5: Thank you for your comments regarding the handling of variables. I have separated the explanatory variables (P5, L179-185) and covariates (P5, L208-210) used in the binary logistic regression analysis into subsections.
Comment 6: This sentence "We also collected age and gender via questionnaire to control for confounding factors affecting fall history" is missing at the end of the text;
Response 6: Thank you for your specific instructions. I have added the suggested content to the section on covariates. Please check P5, L208-210.
Comment 7: In section "2.7. Statistical Analysis" there is a lack of description of the treatment of categorical data;
Response 7: We apologize for the lack of explanation regarding categorical data. Please refer to the additional explanation on categorical data added to P5, L215-218.
Comment 8: The presentation of the analysis procedures contains "logistic regression", however, in the summary, there is an indication of "multivariate analysis". Therefore, confusing: all of this should be rigorously reviewed.
Response 8: Thank you for pointing out the statistical analysis procedure. To avoid misunderstanding, we have standardized the notation used in this study to binomial logistic regression analysis. We have revised the abstract, so please check P1, L19-20.
Results
Comment 9: Again, something confusing: "3.2. Results of binary logistic regression analysis". Where are the results of the multivariate analysis?
Response 9: We apologize for any misunderstanding caused by the description of statistical analysis. In this study, we only used binary logistic regression analysis for multivariate analysis, so we have standardized the description. Therefore, only the results of binary logistic regression analysis are described in “Results.” We appreciate your understanding.
Discussion
Comment 10: At the end of this section, there should be a subsection titled "Clinical Implications." In this subsection, the authors should have presented extensive recommendations for practitioners on how to use their findings in fall prevention;
Response 10: Thank you for your valuable feedback. We have added a subsection titled “Clinical Implications” to the Discussion section. Please refer to P9, L331-338.
Comment 11: Therefore, I suggest creating another subsection titled "Strengths, Limitations, and Future Studies"
Response 11: Similar to “Clinical Implications,” we have added subsections “Limitations” (P9, L340-349) and “Future Studies” (P9, L351-356). Regarding “Strengths,” we found it more coherent to include this information within “Clinical Implications,” so please refer to P9, L331-338.
This is our response to the comments. We appreciate the helpful comments on our paper.
Round 2
Reviewer 2 Report
Comments and Suggestions for Authors
Dear all, I believe that the authors have made corrections to the manuscript and that it is better understood by readers than before.